# An Overview of Serum Albumin Interactions with Biomedical Alloys

**DOI:** 10.3390/ma13214858

**Published:** 2020-10-29

**Authors:** Oksana Klok, Anna Igual Munoz, Stefano Mischler

**Affiliations:** Tribology and Interfacial Chemistry Group, Ecole Polytechnique Fédérale de Lausanne (EPFL), 1015 Lausanne, Switzerland; anna.igualmunoz@epfl.ch (A.I.M.); stefano.mischler@epfl.ch (S.M.)

**Keywords:** biomaterials, metals, corrosion, electrochemistry, metal release, simulated body fluid (SBF)

## Abstract

Understanding the interactions between biomedical alloys and body fluids is of importance for the successful and safe performance of implanted devices. Albumin, as the first protein that comes in contact with an implant surface, can determine the biocompatibility of biomedical alloys. The interaction of albumin with biomedical alloys is a complex process influenced by numerous factors. This literature overview aims at presenting the current understanding of the mechanisms of serum albumin (both Bovine Serum Albumin, BSA, and Human Serum Albumin, HSA) interactions with biomedical alloys, considering only those research works that present a mechanistic description of the involved phenomena. Widely used biomedical alloys, such as 316L steel, CoCrMo and Titanium alloys are specifically addressed in this overview. Considering the literature analysis, four albumin-related phenomena can be distinguished: adsorption, reduction, precipitation, and protein-metal binding. The experimental techniques used to understand and quantify those phenomena are described together with the studied parameters influencing them. The crucial effect of the electrochemical potential on those phenomena is highlighted. The effect of the albumin-related phenomena on corrosion behavior of biomedical materials also is discussed.

## 1. Introduction

Understanding the interaction of biomedical alloys with body fluids is the key factor for the successful and safe performance of the implants in the human body. Indeed, interfacial reactions occurring between biomedical alloys and body fluids determine the durability, corrosion, and wear behavior of artificial implants [1]. Although body fluids are very complex environments, albumin has been reported to be the most abundant protein in the plasma and synovial fluid [2]. Due to its high concentration, albumin comes first into contact with the implant’s surface according to the laws of mass transport, thus, having a significant effect on the initial adsorption of proteins onto surfaces of biomedical alloys [3]. Studies on explants show the presence of protein-containing layers on the surface [3,4], thus highlighting the importance of the understanding of protein interaction with biomedical alloys.

The structure of bovine serum albumin (BSA) (Figure 1) is very similar to the human serum albumin (HSA), which makes it a perfect alternative for HSA in laboratory studies. The protein has a heart-shaped structure, which is maintained by the polypeptide chain (consisting of 585 amino acid residues) folded into three α-helical domains [5]. The dimensions of the BSA molecule are 4 nm × 4 nm × 14 nm [6]. Albumin has a high tendency to reversibly bind to various ligands and metal ions. The literature shows that the interaction of the proteins with biomedical alloys is a very complex process, which depends on the following factors:solution properties: pH [7,8,9], chemical composition and ion concentration (ionic strength) [9,10,11], temperature [3,9,12,13,14];material properties: chemical composition [3,9,15], surface energy [3,16], charge, roughness, wettability [15,17,18,19];protein properties: type [20,21,22,23] and concentration [9,15,24].

Due to the combined effect of these factors, deconvolution of the effect of protein on biomedical alloys remains a challenging task.

Considering that corrosion and tribocorrosion (the combined action of wear and corrosion) phenomena are reported as degradation mechanisms of biomedical alloys in the human body, the impact of albumin on the final corrosion and the tribological response of implants is one of the main motivations for better understanding the interaction of albumin with biomedical alloys. Indeed, the literature reports that albumin affects the corrosion behavior of biomedical alloys by different mechanisms such as accelerating their dissolution [24,26,27,28,29,30,31,32,33] or acting as cathodic inhibitor [34]. Additionally, it also is indicated that albumin might affect the tribological response of the alloys by producing a lubricating effect with the formation of a tribofilm [35,36,37] or changing the viscosity of the fluid and the structure of the double layer [38].

The aim of this literature overview is to present the current understanding of the mechanism of serum albumin (both BSA and HSA) interactions with the most used biomedical alloys (CoCrMo, Titanium alloys, and AISI 316L stainless steel) in simulated body fluids (SBF), focusing on the literature that contains a mechanistic explanation of the different involved phenomena supported by experimental evidences. A literature search is performed on ScienceDirect and Web of Science platforms using the following key words and their various combinations: albumin, protein, BSA, HSA, biomedical, 316L, CoCrMo, and Titanium alloys. The resulting papers from the search are filtered based on the criteria of the mechanistic description of the protein-related phenomena which occur as a result of albumin interaction with biomedical alloys. The final list of documents is presented in Table 1.

Based on those research works, the protein/biomedical alloy interactions can be divided into four albumin-related phenomena: adsorption, reduction, organometallic binding at the surface, or organometallic binding in the solution followed by precipitation. These phenomena can be described as follows:Albumin adsorption phenomenon involves the interaction of albumin and the metallic surface without chemical change of the albumin molecule. This interaction results in the formation of an albumin monolayer or multilayer film either covering the entire metal surface or in the form of islands.Albumin reduction manifests itself in the chemical change of the albumin molecule by the redox reaction at the surface.Albumin binding is the formation of organometallic bonds between an albumin molecule and metallic species at the metal surface.Precipitation occurs when organometallic complexes formed in the solution reach a critical concentration and precipitate on the surface.

This overview describes the four albumin-related phenomena and their impact on the corrosion behavior of biomedical alloys. The experimental techniques used to study these phenomena are reviewed (Figure 2, Figure 3, Figure 4 and Figure 5) and the reported parameters that affect each of the described phenomena are summarized.

## 2. Experimental Techniques Used to Study Interactions of Albumin with Biomedical Alloys

Figure 2, Figure 3, Figure 4 and Figure 5 present the list of the electrochemical (green), microscopic and spectroscopic (red) techniques used for studying the four different protein-related phenomena that occur during interaction of albumin BSA with biomedical alloys. Techniques that allow in-situ investigation of those albumin-related phenomena are marked by (*).

Electrochemical techniques are the experimental techniques used to study in-situ the interaction of proteins with biomedical alloys [58]. Particularly, potentiostatic and potentiodynamic tests are extensively used and useful for understanding and quantifying protein adsorption, precipitation, and reduction [24,29,30,31,34,39,55,56] as these phenomena are influenced by the prevailing potential at the interface.

During a potentiostatic test, a target potential is applied to the metal samples by means of a three-electrode set-up connected to a potentiostat which measures the current at the constant potential as a function of the time in order to follow the electrochemical kinetics of the involved reactions. It allows us to quantify the amount of dissolved metal (faradaic reactions) but, also, to impose different surface conditions (i.e., formation of a certain oxide film by imposing an appropriate potential within the passive domain of a CoCrMo alloy or stainless steel). Chronocoulometry and coulometry are electrochemical techniques used for the determination of the faradaic charge for BSA reduction [49] and can be considered a specific use of the potentiostatic technique. More specifically, potentiostatic coulometry is an electrochemical technique in which the constant potential is applied to the working electrode while the current that flows through the circuit is measured. Applying a constant potential for a sufficient amount of time allows it to fully oxidize or reduce all of the electroactive species in the solution (i.e., BSA). As the electroactive molecules are consumed, the current also decreases and approaches zero when the conversion is complete. During the chronocoulometric technique, the entire charge that passes under the application of a step potential is recorded as a function of time. Integration of the current through the applied potential allows us to calculate the charge [59].

Potentiodynamic tests allow us to measure the current density as a function of the applied potential, which is swept at a constant scan rate by applying a function generator to drive the potentiostat. Potentiodynamic tests allow us to evaluate the effect of BSA on the different electrochemical reactions, which occur depending on the potential. However, potentiostatic and potentiodynamic tests do not allow for the quantitative estimation of the adsorbed or precipitated organic matter, as well as for the tracing of the kinetics of these processes, and complementary techniques are typically used combined with them (i.e., surface analysis or metal ion analysis in the solution).

QCM or its electrochemical version EQCM, combined with the electrochemical cell, is a highly-sensitive technique that allows us to determine the mass changes in the nanogram range by measuring the variation of the resonant frequency of a piezoelectric material. Any variation in the mass, which is firmly attached to the working electrode, is reflected in an alteration of the quartz crystal’s resonance frequency [60]. Thus, the shift of the resonance frequency provides a quantitative estimation of the mass change of the working electrode. It has been successfully used to study the adsorption and precipitation phenomena of BSA on different biomedical alloys [34,39,40,41,55,57]. The reversibility of protein adsorption also is determined using the EQCM technique with a flow-cell system which allows us to change, in a controlled way, the chemistry of the solution (i.e., injecting proteins through a fluidic system) [61].

CV is another direct current (DC) electrochemical technique consisting of the registration of the current response to a potential that varies at a constant scan rate in the forward and reverse directions, during one or more cycles [62]. It is used to study the adsorption and reduction phenomena of proteins and to determine their reversibility [42,43,49]. It also is used to quantify the charge density involved in the protein adsorption [42,43].

EIS is an alternating current (AC) electrochemical and non-destructive technique (when used at OCP conditions) to study the interaction of proteins with biomedical alloys. The EIS method involves measuring the response of the working electrode to a low amplitude (approximately 5–10 mV) sinusoidal potential modulation at various frequencies [63]. EIS is sensitive to small changes in the system, which gives a possibility to characterize the material properties and electrochemical systems even in low conductivity environments. One common way for the interpretation of the impedance data involves the selection of an equivalent electric circuit that fits the experimental results to a combination of electrical elements (resistance, conductance, and inductance elements), which must be defined according to the physical and electrochemical description of the studied system. EIS is used to identify the resistance and capacitance of the precipitated or adsorbed protein layer [44]. Additionally, EIS allows us to determine the adsorption mechanisms and corrosion kinetics during the protein adsorption by carrying out electrochemical tests at different temperatures and protein concentrations, for example. Indeed, the activation energy for BSA adsorption is quantified under those variable conditions using EIS [62].

It is important to highlight that some of the protein/metal interactions involve non-electrochemical reactions (i.e., protein binding to metal ions and precipitation of the formed organometallic complexes) and, therefore, the electrochemical techniques are insufficient to describe them. Therefore, ex-situ experimental techniques or in-situ analytical techniques (i.e., inductively coupled plasma optical emission spectrometry) constitute alternative and also complementary experimental tools.

Ex-situ material analysis techniques are frequently used in combination with electrochemical techniques, providing complementary information in the understanding of the mechanisms involved in the interaction of proteins with biomedical alloys. Surface topography and morphology can be assessed using SEM and AFM. Additionally, AFM is helpful in understanding and quantifying the thickness, compactness, and the binding forces of the protein layer to the metallic surface [44]. Apart from the ex-situ studies [44,50,64], AFM technique also is reported to be used in-situ during electrochemical tests [45]. In Ref. [55], the adsorption of BSA as a function of pH and surface charge of a CoCrMo alloy was studied by in-situ AFM and the authors were able to measure an increase in corrosion of the alloy promoted by the BSA adsorption.

EDS and WDS are used for qualitative and quantitative chemical analysis of the surface oxide and organic layers [26,27,28,29,30,34,44,46,47,51,52,54,55], while XPS and AES additionally provide depth resolution, thus information on the thickness and chemical composition of these layers [29]. The influence of protein adsorption on the surface chemistry of CoCrMo alloys have been studied combining in-situ electrochemical techniques and ex-situ surface analysis [29,34]. Those studies have provided an understanding of the effect of BSA on the passive dissolution of CoCrMo alloys.

To identify different properties (chemical or conformational) of the adsorbed organic substances, specific surface analysis techniques such as FT-IR, RA-IR or FT-IRRAS are used. They allow us to identify the structural and conformational changes of the adsorbed or bonded protein based on the ratio of α-helix, the β-sheet, and the disordered structures, which is reflected by the spectral regions of amide I, II and II bands [15,48]. XRD also is reported to be used to determine the conformational structure of the protein layer [46]. The inability to perform the in-situ analysis is a main drawback of the majority of the surface analysis techniques because ex-situ analysis might introduce changes in the surface chemistry, for example, due to cleaning procedures and contact with air when the samples are transferred into the chambers for analysis [65]. Additionally, high vacuum techniques might be demanding in terms of preserving the protein structure [15]. Alternatively, ellipsometry is an in-situ technique which allows us to study the protein adsorption kinetics and to quantify the thickness and density of the adsorbed protein. However, identification of the configuration, orientation, and chemistry of the adsorbed protein is not possible with this technique [15].

Apart from the analysis of the surface, ICP-OES and AAS are typically used analytical techniques to measure in-situ the non-precipitated amount of the released metal in the solution [26,27,51,52,54]. The output obtained from these techniques, metal ion concentration, as a function of time gives information on the kinetics of metal ion release. However, these techniques do not allow us to access the real amount of released metal ions in the case of their complexation with a protein or precipitation of the organic complexes. Therefore, the digestion of the solution prior to the analysis is needed. Additionally, the method of solution sampling may affect the results. It was shown that pipetting solution from the top of the experimental setup results in a lower measured amount of metal ions compared to those cases where the whole test solution is tested [26]. This difference is associated with the tendency of the protein-metal aggregates to accumulate at the bottom of the experimental setup.

## 3. Albumin Adsorption

Adsorption of proteins in biological fluids is one of the initial steps that affects the biocompatibility of the implants inside a human body [66]. BSA adsorption was observed and corroborated either by in-situ electrochemical and/or ex-situ surface analysis techniques by different authors [24,26,29,30,34,39,42,43,44,46,50,67,68,69,70]. Indeed, several reviews dedicated to adsorption phenomena involving various proteins exist in the literature [3,9,15,71]. Considering the thermodynamic perspective, the entropy gain is reported to be a driving force for BSA adsorption [9] on stainless steel [72], CoCrMo [30] and Ti [42]. The gain in entropy gives a clear indication of conformational changes of adsorbed albumin, which could be caused by protein structural unfolding upon adsorption, compared to a protein from the solution. However, there are no sufficient evidences to claim that the entropy gain is the only driving force for the adsorption, as the latter also is affected by the surface energy [3,9,15,16,71], surface charge [3,9,15,45], and electron conductivity [15,73]. Furthermore, since protein adsorption is a dynamic process, the driving force may change with time, in a row of subsequent events, which makes it difficult to evaluate separately the impact of each surface characteristic. Therefore, a comprehensive evaluation of the physicochemical properties of the proteins and surfaces is quite important for the understanding of the adsorption phenomenon and its correlation with the surface characteristics [15].

Taking the mechanistic point of view, in the literature, albumin adsorption is reported to occur by chemisorption [3,24,30,42,43] or through electrostatic interactions [3], while covalent bonding also is suggested [41]. Still, there are some crucial open questions in this field, mainly related to the kinetics of BSA adsorption on different surfaces. This Section aims at providing the published mechanistic insights on the effect of the different studied parameters (surface chemistry, protein concentration, pH, surface charge, solution chemistry) on the albumin adsorption phenomenon. The summary of the parameters, which affect the albumin adsorption and other albumin-related phenomena considered in this overview, is given in Section 7.

### 3.1. Effect of the Surface Chemistry of the Alloy

The effect of the surface chemical composition on the adsorption of BSA on single and binary-component surfaces is systematically studied in the literature [47,74,75]. These studies show that protein adsorption strongly depends on the composition of native oxide, while no obvious correlation with the bulk composition of the alloy is observed. Furthermore, the surface chemical composition affects conformational changes in the BSA molecule upon adsorption. The effect of surface chemistry on adsorption is mainly rationalized in terms of changes on surface energy and surface charge.

Among the studied single metals, the lowest adsorption is observed on Al, which is covered by the native Al_2_O_3_ oxide, while the BSA adsorption increases in the following sequence: Al, Ni, Cr and Ti [47]. Variation in BSA adsorption on Ti and on Cr is within the statistical error. During all cases, adsorption of the albumin monolayer is reported to occur via a side-on orientation (long axis of protein oriented parallel to the surface). Ref. [48] shows that in an aqueous solution, BSA adsorbed on a chromium surface is more compact compared to molybdenum. This difference is attributed to the observation that chromium is completely covered by a hydroxide film, while molybdenum is only slightly hydroxylated. This difference suggests that upon adsorption the BSA structure reorganizes to facilitate its interaction with the hydrophilic Cr surface. The following mechanism of sequential BSA adsorption is suggested to explain the difference of BSA adsorption of Cr and Mo [48]. Upon the contact of the negatively charged BSA molecule with the surface, it partially unfolds to direct a maximum amount of positive groups, such as the N-terminal and lysine residue, toward the metallic surface. This process occurs on both Cr and Mo within the initial 20 min of exposure to the BSA-containing solution. Subsequently, the adsorption of proteins proceeds, further minimizing the surface energy in two possible scenarios. Regarding Mo, BSA occupies the free surface areas. The other scenario involves the growth of a second layer of proteins on the already attached ones. The proteins in the second layer are suggested to be less unfolded than the ones that are directly attached to the surface and they may possess a more compact helical structure. This mechanism is probably predominant on Cr, as suggested by FT-IRRAS results [48]. Nevertheless, this mechanism is based on the assumption that at a pH of 7–8.3, both surfaces are negatively charged; however, in the recent review by Kosmulski presented in Ref. [76], the Cr_2_O_3_ oxide was reported to have zero charge in this pH range. Therefore, the influence of surface chemistry on the adsorption mechanisms of BSA is not clear yet.

Concerning the case of binary-component metal surfaces, it is found that protein adsorption always decreases in the presence of Al_2_O_3_ in the various binary films, and the higher the fraction of Al oxide in the film, the lower the protein adsorption [47,74,75].

In Ref. [16], it was found that the influence of oxide chemical composition on adsorption is related to its influence on surface energy. The authors observed that the decrease of the polar components on the surface, accompanied by the increase in the surface hydrophilicity, reduces the amount of adsorbed protein on thermally treated CoCrMo alloys. Co and Cr were found to have greater polarity than CoO, and the polarity of CoO was greater than the polarity of Cr_2_O_3_. This result shows that BSA adsorption on a CoCrMo surface is influenced by electrostatic and hydrophobic interaction.

The increase of the albumin and fibroconnectin adsorption with an increase of the polar component of the surface energy induced by oxidation also is reported for NiTi alloy, which remains hydrophilic [15,77]. It is suggested that adsorption of albumin on slightly hydrophilic surfaces like TiO_x_ is largely affected by electrostatic interactions [78].

The effect of the surface chemical composition on albumin adsorption also is associated with the electron permittivity of the surface oxide [15,73]. Studying the platelet adhesion to Ti, Ti–6Al–4V, Ti–6Al–7Nb, 316L steel, and CoCrMo alloy, in Ref. [73] it was suggested that a larger albumin adsorption on 316L and CoCrMo might be associated with the lower relative permittivity of the native Cr oxide compared to Ti oxide, which allows Cr_2_O_3_ to maintain larger electrostatic forces. Therefore, the adsorbed albumin would play a role of inhibitor for platelet adhesion and clot formation.

Thus, the surface chemical composition of the alloy affects the albumin adsorption. Attempts to correlate the albumin adsorption with the alloy chemical composition in terms of the effect of the latter on the surface energy, polarity, surface charge and hydrophobicity have been done in the existing literature. Since these properties might be interrelated, it remains difficult to distinguish whether this is an effect of a single parameter or their combination.

### 3.2. Effect of Protein Concentration

Protein concentrations ranging from 0.05 to 30 g/L were used by different authors when studying the effect of protein concentration on its adsorption. The choice of those amounts is based on the values typically found in synovial fluids (around 30 g/L) and much lower ones for developing a fundamental understanding of the adsorption mechanism. The increase of BSA concentration from 0.05 to 0.5 g/L is reported to increase protein adsorption on CoCrMo [24], while BSA concentration from 15 to 30 g/L is reported to result in thicker protein layers on CoCrMo [44]. Contrary to results from Ref. [24,44], no further increase in HSA adsorption on Cr at concentrations exceeding 0.5 g/L HSA was observed in Ref. [50] after reaching a mass corresponding to one full monolayer coverage.

In Ref. [44], the effect of clinically relevant protein concentrations (15–30 g/L) on the surface status of a CoCrMo alloy at cathodic potentials −0.7 V_SCE_ and −0.9 V_SCE_ and temperatures of 37 °C and 50 °C in PBS solution was investigated. It was found that at the highest BSA concentration, 30 g/L, and cathodic potential, −0.9 V_SCE_, the resistance and resistivity of the oxide were the lowest at both temperatures, and this effect was the most pronounced at 50 °C. Generally, the oxide resistance and resistivity are always negatively influenced by cathodic polarization to −0.9 V_SCE_ for all tested protein contents and temperature ranges when compared to −0.7 V_SCE_. Based on that, the decrease in resistivity of the oxide at −0.9 V_SCE_ is associated with the applied cathodic potential, which results in depletion of Cr in the surface oxide underneath the absorbed protein observed by TEM. Based on the AFM results, the protein layer formed on the surface at the highest BSA concentration and temperature is the thickest and most compact, and possesses the least adhesion to the surface, compared to the layer formed at lower temperatures or BSA concentration. The effect of the albumin concentration was explained by the authors as follows: at a low protein concentration, 15 g/L, the protein adsorption is slow which allows conformational and orientational changes in protein molecules to occur during adsorption. This leads to the increased adhesion of the adsorbed layer to the surface. Occurring at a high protein concentration, 30 g/L, the adsorption process at the surface proceeds faster, which allows us to reduce the structural changes in the protein molecule and results in the formation of a saturated adsorbed layer. Moreover, the configuration of the adsorbed protein can be affected by bulk protein concentration expressed by ionic strength. The high ionic strength promotes formation of a compact layer due to shielding some of the repulsive forces between molecules, while low ionic strength results in the formation of loosely assembled proteinaceous film [44].

The effect of BSA concentration (from 10 μg/mL to 1 mg/mL) and applied potential on protein adsorption kinetics on Ti in PBS was investigated in Ref. [39], using EQCM. The authors observed a continuous decrease in mass of 420 ng/cm^2^ and 80 ng/cm^2^ in the 1 mg/mL and 10 μg/mL BSA-containing solutions, respectively, at an applied cathodic potential of −1.2 V_Ag/AgCl_. Since no clear mass loss is observed on Ti in pure PBS (Figure 3), the mass reduction is attributed to desorption of proteins from the surface. This protein desorption at −1.2 V_Ag/AgCl_ is suggested to be a result of electrostatic repulsion between the negatively charged BSA at pH 7 and the negatively charged titanium surface combined with the hydrogen generation during water reduction.

Occurring at an applied anodic potential 1 V_Ag/AgCl_, the measured mass change is observed to be dependent on the BSA concentration: no mass change is observed in PBS containing 1 mg/mL BSA, while a continuous increase of mass is reported in a 10 μg/mL BSA solution, giving a final mass gain of approximately 305 ng/cm^2^ (Figure 6). The authors attributed this difference to the sequence of their experimental procedure in which the QCM sensor surface already could be saturated with protein before the application of the anodic potential. Thus, this work (Figure 2) shows that at cathodic potential, desorption of BSA from the Ti surface occurs due to the protein dehydration triggered by water reduction and electrostatic repulsion, while anodic potentials promote BSA adsorption only when the Ti surface is not saturated already with protein [39]. Clearly, the effect of BSA concentration on its adsorption is dependent on the surface properties.

The effect of protein concentration on adsorption and its consequences on the corrosion behavior of biomedical alloys are of special relevance and studied by several authors [24,39,44]. In Ref. [24], for example, the corrosion behavior of a CoCrMo biomedical alloy at different temperatures, 25, 40, 50 and 60 °C in different BSA concentrations (0.05 and 0.5 g/L of BSA) was analyzed. The authors found that BSA inhibits the cathodic reaction independently of the temperature and this behaviour is more pronounced at low BSA concentrations. Conversely, BSA catalyzes the anodic reaction, but this effect also is influenced by the BSA concentration and solution temperature. Specifically, the increase in BSA concentration to 0.5 g/L slightly decreases the anodic dissolution in comparison with a lower concentration of 0.05 g/L at 25–40 °C, which is attributed to a larger adsorption of BSA at higher concentrations, impeding the mass transport from the electrode surface.

Applying the Langmuir isotherm model, the saturated surface concentration, Γ_max_, is found to increase with temperature, while the affinity of the adsorbate toward adsorption, B_ads_, decreases between 25 and 50 °C and then increases at 60 °C. Since BSA denaturation begins at 60 °C [79], the higher affinity of the denatured BSA molecule is suggested to be responsible for the increase of BSA adsorption, possibly resulting in multilayers [24].

Thus, based on the literature, the effect of BSA concentration on its adsorption on the considered biomedical alloys is ambiguous. It is difficult to generalize the existing data and to deconvolute the pure effect of the protein concentration on protein adsorption, as in most of the studies the effect of protein concentration was investigated in combination with other parameters, such as temperature or applied potential. Based on that, it can be concluded that a systematic study on the effect of BSA concentration on adsorption phenomena is still missing. Particularly, the quantitative evaluation of the adsorbed protein in-situ is essential. Additionally, the use of the flow cells with injection loops is recommended to reduce the experimental uncertainty due to the pre-adsorption of albumin on the tested metallic surfaces prior to the experiments. The effect of protein concentration on conformational changes on the considered alloys also has to be verified.

### 3.3. Effect of pH and BSA and Surface Charge

A variation of the pH of the electrolyte affects the net charge of BSA. The isoelectric point (IEP) of the BSA is 4.7–4.9 in water at 25 °C [80]. Thus, for a pH lower than the IEP the protein has a positive charge, while for higher pH values, the protein charge is negative. The mechanism of the effect of pH on the BSA charge is described by Ref. [45] as follows: when pH decreases, H^+^ binds to the COO^−^ groups, which neutralizes the negative charge of the latter. Additionally, H^+^ binds to the unoccupied pair of electrons on the N atom of the −NH_2_ amino groups, which changes their charge to positive. Consequently, the net charge of the BSA molecule becomes more positive and –R groups tend to show ionic (electrostatic) interaction with other molecules. When pH increases, H^+^ are eliminated from the COOH groups, turning them into the negatively charged COO^−^, while removal of H^+^ from the NH^3+^ groups eliminates their positive charge, which alters the BSA net charge to more negative [45]. The BSA charge may affect its interaction with the surface, which charge also is influenced by the electrolyte pH.

Thus, the effect of the solution pH on BSA adsorption is related to the charge of the protein molecule and the relation between the charge of the BSA molecule and metallic surface. The ranges of the literature-reported IEP data for BSA [80], Cr_2_O_3_ and TiO_2_ [76] are summarized in Figure 7. The relation between the BSA charge and surface also can be affected by the externally applied potential, which changes the surface charge, as shown in Ref. [45]. The authors show that variation of pH at OCP does not affect the protein adsorption and the hydrophobic interaction is responsible for adsorption forming a monolayer. Regarding the positively charged CoCrMo surface (at 0.6 V_Ag/AgCl_), the highest adsorption is observed at IEP of BSA (pH 4.7), while deviating from this value to pH 3 and 10 BSA adsorption is reduced, especially in the acidic solution where both surface and protein are positively charged. This suggests that electrostatic interactions are responsible for the decrease in adsorption. Concerning the case of the negative surface charge (−0.8 V_Ag/AgCl_), the pH 4.7 and 10 could affect the net charge of the BSA molecule, reducing the adsorption due to electrostatic repulsion, which also can be the possible reason for the decrease of BSA surface coverage obtained by Ref. [41] on the Cr in an Na_2_SO_4_ supporting solution with pH 5.5 and pH 10 at passive potentials. However, contrary to the results from Ref. [41] obtained at pH 4, no enhancement of the protein adsorption was observed by Ref. [45] at pH 3 and passive negative potential, despite the opposite charge of the surface and BSA, which would be expected to increase electrostatic attraction. This effect could be associated with the competitive adsorption of phosphates and BSA at this pH [30,31] and it will be discussed more in detail in the following Subsection.

The relation between the charges of the surface and BSA also can explain the higher adsorption of BSA at 0.35 V_Ag/AgCl_ compared to −0.05 V_Ag/AgCl_ in a PBS solution at pH 7.4 in Ref. [67], as at this pH BSA is negatively charged and, with the increase in the potential, the surface charge becomes more positive, inducing the electrostatic attraction for BSA adsorption.

### 3.4. Effect of Phosphates

The pH in biological systems is maintained mainly by a phosphate buffer at 7.4 [81]. Phosphate buffer solutions are commonly used as a base electrolyte for BSA adsorption studies to keep the solutions buffered at pH 7.4, although their phosphate concentration is higher than the amount reported in blood plasma [82] and synovial fluids [83]. Therefore, the combined effect of the phosphates and BSA on corrosion of biomedical alloys was studied in literature [29,30,31].

In Ref. [29], the interactive effect of phosphates and albumin on the passive behavior of CoCrMo at pH 7.4 in model SBF was investigated. The authors found that there was a competitive adsorption between the phosphate ions and the BSA molecules, favored at pH 7.4 where both albumin and phosphates are negatively charged. Concerning the effect of phosphates and BSA adsorption on the corrosion behavior of the CoCrMo alloy, it was observed that in pure PBS solution phosphates increase the corrosion resistance, while the presence of albumin decreases corrosion resistance in a PBS+BSA solution. This effect was described using the Evans diagrams shown in Figure 8 [29]. Based on the mixed potential theory, albumin plays an antagonistic role, i.e., promotes metal dissolution, while at the same time decreases the cathodic current by blocking the access of the oxidant to the surface. These two effects balance each other, to a certain extent, close to corrosion potential. Moving to more anodic potentials, the impact of the cathodic current becomes lower and corrosion is significantly accelerated by albumin. Phosphate adsorption tends to decrease the anodic reaction. During the presence of both species in solution, the competitive adsorption of albumin replaces adsorbed phosphate ions and results in acceleration of the anodic reaction.

It was shown in Ref. [30,31] that in PBS solution BSA affects the electrochemical behaviour of a CoCrMo alloy depending on the pH: it has a moderate effect in acidic solutions (pH 3), while in neutral (pH 7.4) and alkaline (pH 10) solutions its effect on the cathodic and anodic reactions is more pronounced. This behavior again is attributed to the competitive adsorption of BSA and phosphate compounds, which exist in solutions at different pHs. Depending on the nature of those phosphate compounds, the BSA either enhances or blocks the passive dissolution. Occurring at pH 3, BSA addition decreases the resistance of CoCrMo to passive dissolution due to substitution of H_2_PO_4_^–^ sites, which could block electronic transfer through the passive film and diminish passive dissolution. However, the effect of BSA at pH 3 is not pronounced, as the BSA has a positive charge, while phosphates are negatively charged and therefore, predominantly adsorbed on the surface. Thus, at pH 3 the adsorption of BSA is the lowest and the H_2_PO_4_^−^ concentration is the highest, which results in the improved corrosion resistance of the CoCrMo under passive conditions. However, BSA adsorption slightly improves the resistance to passive dissolution at pHs 7.4 and 10 when substituting HPO_4_^−2^ sites, which do not have a strong blocking effect against the passive dissolution of the CoCrMo alloy [30].

Competitive adsorption of calf serum proteins and phosphates also is reported for Ti [84]. In Ref. [84], the decrease in protein adsorption on the titanium surface was attributed to the change in the surface wettability in the presence of phosphates.

Regarding the case of 316L stainless steel, the XPS results obtained in Ref. [51,54], on the surface film of 316L steel exposed for 22 weeks at 37 °C to PBS+BSA and PBS + HSA solutions, in comparison with pure PBS, showed a decrease in P surface concentration and this decrease was more pronounced with an increase of both protein concentrations. In Ref. [40], the inhibition of an anodic reaction on 316L steel in a PBS solution was reported, while this effect was diminished in the presence of BSA resulting in an increase of anodic dissolution.

Contrary to the phosphates, which compete with BSA for adsorption onto the metal surface, other solution species might facilitate the protein adsorption. Specifically, it is suggested that Ca^2+^ from solution favors the BSA adsorption on CoCrMo acting as an intermediate bridge, which introduces a positive charge to the surface and electrostatically attracts the negatively charged RCOO^−^ groups of BSA [46]. The similar effect of Ca^2+^ acting as a bridge for BSA adsorption also is mentioned for Ti–6Al–4V [85].

### 3.5. Adsorption Models

Protein adsorption equilibria can be described using the adsorption isotherms. The Langmuir isotherm is reported to be used for the description of BSA adsorption onto surfaces of metallic materials considered in this overview [29,42,72,81,86]. The Langmuir model corresponds to a dynamic adsorption process with the exchange of the adsorbed and the free proteins when there is no intermolecular interaction between the adsorbed molecules. Using the Langmuir model, a monolayer coverage of adsorbate over a homogeneous adsorbent surface is assumed. The Langmuir model can be expressed by the following equation:(1)$$\Gamma =\frac{B_{\mathrm{ADS}\;}\Gamma_{\max\;}c}{1+B_{\mathrm{ADS}\;}c}$$
where Γ  is the amount of the protein adsorbed, (mol cm^−2^); $B_{\mathrm{ADS}\;}\mathrm{is}$ the affinity of the protein molecules toward an adsorbent surface, (cm^3^ mol^−1^); $\Gamma_{\max\;}\mathrm{is}\;$ the maximum value of Γ (mol cm^−2^); *c* is the equilibrium concentration of the adsorbate (BSA) in the bulk solution (mol cm^−3^). The Langmuir model is applied to describe adsorption of BSA on 316l steel [72], CoCrMo [29,81] and Ti [42,86].

Application of the aforementioned model allows us to extract various thermodynamic parameters which provide an understanding of the adsorption process. Specifically, B_ADS_ reflects the affinity of the adsorbate molecules toward the studied surface at a constant temperature. The Gibbs free energy of adsorption, ΔG_ADS_*,* represents the energy involved in the adsorption process and the spontaneity of the adsorption (spontaneous if negative). The value of enthalpy, ΔH_ADS_*,* allows us to distinguish if the process is endothermic (positive) or exothermic and the mechanism of the protein adsorption (i.e., chemisorption for ΔH_ADS_ values around −20 kJ mol^−1^). Comparing the values of enthalpy ΔH_ADS_ and entropy ΔS_ADS_, one may define the driving force for adsorption. The higher value of the ΔS_ADS_ term indicates that process is entropically governed and results in changes of the structure and conformation of the protein molecule upon adsorption [62].

Despite the current progress in the fundamental understanding of protein adsorption, due to the overall complexity of this process there is still a lack of reliable and validated long-term kinetical adsorption models for various biomedical alloys. Use of sensitive techniques such as EQCM in combination with surface analysis techniques might be helpful to understand the kinetics and the chemical nature of the processes that occur during adsorption of proteins onto metallic surfaces. Since ex-situ surface analysis might introduce changes in surface chemistry due to contact with ambient atmosphere or changes in the thickness of the adsorbed layer due to cleaning procedures [65], a possibility of performing in-situ chemical characterization should be prioritized.

## 4. Albumin Reduction

Albumin reduction and its effect on the corrosion of biomedical alloys is one of the least described phenomena, which is probably associated with the complexity of the BSA molecule.

The evidences of BSA electrochemical reduction are demonstrated in Ref. [49]. According to Ref. [49], the reduction of albumin seems to be a prerequisite for its adsorption. The applied potentials below −0.6 V_Ag/AgCl_ for times shorter than 1 h was suggested to lead to the electrochemical reduction of 3 or 4 of the 17 disulfide bonds in the adsorbed BSA molecule on an Hg electrode. Longer exposure time increases the number of reduced disulfide bonds up to 8 or 9. After reduction, the BSA forms an insoluble product which is strongly adhered to the electrode surface.

According to the model proposed in Ref. [49], the electrochemical behavior of the BSA adsorption process implies adsorption as the result of interaction of disulfide bonds with the surface of an Hg electrode. This interaction results in the formation of SH-groups on the reduced bonds. Meanwhile, the BSA molecule remains at the electrode surface and its structure is maintained by intramolecular hydrogen bonding and internal disulfide bonds.

Electrochemical reduction of BSA is suggested to be a reason for the increased albumin adsorption on a CoCrMo alloy at cathodic potentials in a 0.14M NaCl + BSA solution observed in Ref. [34]. However, the direct BSA reduction cannot explain the obtained higher values of the measured cathodic current compared to the estimated ones for the process governed by BSA reduction. It is suggested that the BSA reduction under cathodic potentials in aqueous solutions occurs in two steps: initial hydrogen evolution by water reduction (Reaction (1)) and subsequent bonding of hydrogen to the disulfide groups of BSA (Reaction (2)):

(1) H_2_O + 2^e−^ → 2H_ADS_ + OH^−^

(2) RSSR + 2H_ADS_ → RSHHSR_ADS_

## 5. Albumin-Metal Binding

Protein-metal binding manifests itself in a formation of organometallic complexes at the surface. Typically, an organometallic complex consists of two moieties: organic and metallic. Depending on the interaction between these moieties, it can be chelation, coordination, or ligand formation [87]. Protein can bind to metal atoms on the metal surface oxide or to dissolved metal ions in the solution.

Binding of the protein to metal atoms depends significantly on the properties of the oxide surface, and it is only possible if the protein-metal bond is stronger than metal-oxide or metal-hydroxide ones, or when these bonds became weak due to the presence of defects. Typically, weakening of the metal bonds is slow. Once weakened, the protein-metal complex detaches from the surface. The detachment of the protein-metal complexes progresses in a slower manner than the protein-metal binding [17]. Protein binding to metal, and detachment of the formed complexes, depend on factors such as composition of the alloy and surface oxide [27,51,54], heterogeneity, crystallinity [88,89], and the density of the defects of the latter [17,52]. Specifically, amorphous oxide seems to facilitate metal dissolution resulting from the protein-metal interaction [88,89]. This effect possibly occurs due to a weaker bond between metal and oxide than in crystalline oxides [17]. Additionally, metal release induced by protein-metal binding also depends on other parameters, such as adsorption mechanisms, metal atom availability in the surface oxide, stability constants of protein surface groups and metal ions [3,90], protein concentration [91] and the temperature of the solution [26]. Albumin was reported to bind to Fe [28,54], Cr [27,91,92,93,94], Ni [26,51], Co [26,27], Mo [54], Ti [51,54], Al [95,96,97], and V [53].

Ref. [26] suggests the mechanism based on complexation/ligand-binding processes, which is controlled by the adsorption and governs the metal release from 316L stainless steel. Thus, proteins initially adsorb on the steel surface, reducing OCP values. Further, the adsorbed protein undergoes complexation with the surface oxide/hydroxide metal atom followed by the detachment of these complexes into solution [90]. Detachment of the protein-metal complexes may be promoted by the presence of other protein types. i.e., the Vroman effect [98]. The Vroman effect describes the phenomena when the proteins adsorbed at the surface are displaced by another type of protein of a larger binding affinity, typically a larger sized one. The increase in the dissolution of Fe, Cr, Ni from 316L after the addition of fibrinogen to a PBS + BSA solution was clearly observed in Ref. [26].

The highest amount of Fe release from the 316L stainless steel [28,51,54] can be explained by its highest concentration in the alloy and the structure of the passive layer. The passive layer of 316L stainless steel contains a Cr-rich compact inner oxide and an outer Fe oxide, which less compactly bonds to the surface [72]. Thus, the iron dissolution is more pronounced compared to other alloying elements in 316L steel. Fe is reported to have an affinity to form soluble components with albumin. Regarding albumin-containing solutions, Fe_2_O_3_ and FeOOH are leached out as a result of the interaction of Fe with negatively-charged carboxylate groups of proteins with a subsequent formation of protein-Fe complexes [28]. An increase in Fe ion dissolution in the presence of albumin, and with the increase in albumin concentration, also supports the affinity of Fe to interact with this protein [26,28,51,54]. The release of the Fe from the surface oxide is the rate-limiting step in dissolution induced by the complexation/ligand binding [12] which causes an enrichment in Cr content in the oxide layer.

According to Ref. [52], a complete monolayer coverage of adsorbed BSA is needed to both significantly increase the metal release and to promote subsequent Cr enrichment in the surface oxide compared to a protein-free solution. Approximately one monolayer of adsorbed protein on 316L steel in a PBS solution with 1 g/L BSA is reported to occur within 1 h [52]. An increase in the protein concentration from 1 g/L to 100 g/L in long-term tests (168 h) is found to increase the metal release without a significant change to the thickness and structure of the adsorbed albumin layer indicating that, except for the complexing effect of adsorbed proteins, solution proteins and/or the exchange of adsorbed ones are important effects that also influence metal release [12,52]. According to Ref. [52], the BSA concentration of 1 g/L seemed to be a threshold value to induce a profound metal release and chromium enrichment in surface oxide, since no significant effect of BSA was observed at concentrations of 0.1 and 0.01 g/L in comparison to BSA-free solutions.

Considering the Cr release from 316L steel in the presence of albumin, the literature data are contradictory: some of the studies report an increase in Cr dissolution in an albumin-containing solution [12,26,28,52], while no obvious effect is observed in Ref. [51,54], suggesting that Cr does not seem to remarkably interact with albumin. Complexation studies in Ref. [91] show that Cr(III) forms weak complexes with proteins in solution, and this process is strongly affected by the protein size and concentration. The higher concentration of albumin in PBS solutions [12,26,52], possibly can be responsible for the enhanced dissolution of Cr compared to Ref. [51,54].

Concerning the Ni release from 316L steel in the presence of albumin, no clear effect is reported by Ref. [28]. This effect is explained by the presence of the oxide film, which prevents the direct access of the adsorbed albumin to nickel, which exists as metallic nickel enriched in the substrate under surface oxide [28,99]. A slight increase of the Ni release in the presence of albumin is reported in [12,26,54]. Release of Ni is especially promoted in the presence of HSA [51], which is associated with the high affinity of HSA to bind with Ni through N-terminal binding sites. The presence of defects in the oxide film may facilitate the access of protein to the Ni-rich layer underneath [52]. Mn release is shown to be the least affected by complexation mechanisms, but mostly by the lowering of the solution pH by BSA [52].

Regarding a CoCrMo alloy, it was observed in Ref. [27] that in a PBS+BSA solution, an initial increase of the Co atoms released per surface area is followed by a decrease after 1 and 4 weeks of exposure, while the concentration of Cr and Mo increases. The decrease in the Co concentration in the surface oxide confirmed by XPS is only partially responsible for this effect. Based on the underestimation of the aqueous Co in PBS+BSA solution samples studied by AAS, it is postulated that Co preferentially binds to protein compared to Cr and Mo, leading to the protein aggregation and subsequent precipitation from the solution. Ref. [54] reports the increase in the Mo release in a PBS+BSA solution after 14 weeks, and it is more substantial when increasing the BSA concentration from 0.2 to 4 g/L. However, the released amount of Mo decreases after 22 weeks in BSA-containing solutions compared to a BSA-free one, indicating the binding of Mo to BSA and subsequent precipitation.

This study also reports the tendency of Mo to bind to BSA forming organometallic complexes, which precipitate on the surface rather than dissolve in the solution, which is confirmed by XPS results [54].

Exposure of Ti–6Al–4V to a PBS+BSA solution shows no effect of BSA on Ti release; however, XPS results [51,54] suggest that Na adsorbs on the alloy surface and acts as a bridge to bind the sulfhydryl group of BSA to Ti. Regarding the HSA solution, Na is suggested to bridge the Ti–6Al–4V surface via the carboxyl group of HSA. V release is impeded in the presence of both BSA [54] and HSA [51], as compared to a PBS solution, which is explained by the formation of an adsorbed protein layer [51]. Although it is reported that V tends to form BSA/VO_3_^−^ and BSA/VO_2_^+^ complexes [53], as well as HSA/VO_2_^+^ [100], the concentration of released V increases with time, indicating that precipitation of such complexes is unlikely [51]. This behavior indicates that interaction of proteins with alloys may differ from the interaction with metal salts. This also concerns nanoparticles and powders, as they may have totally different surface properties compared to the bulk materials [87].

The enhanced metal release from a Ti–6Al–4V alloy in the presence of BSA and hydrogen peroxide (H_2_O_2_) was observed by Ref. [64,80]. The increase of metal release is more sensitive to the concertation of H_2_O_2_ than BSA [80], which suggests that H_2_O_2_ is mainly responsible for the observed effect due to its complexation with Ti from the surface oxide. The surface oxide exhibits enrichment of Al, however, its fraction reduces with increasing BSA concentration, which is possibly caused by a delay of interfacial reactions due to albumin adsorption or by complexation of BSA with Al. Similar enhancement of metal release in the presence of BSA and H_2_O_2_ is reported for 316L steel [28]. BSA is found to decrease the release of Ni enhanced by H_2_O_2_, while facilitating the dissolution of the Fe and Cr oxides induced by H_2_O_2_ and leading to a profound Fe and Cr release as metal ions and/or protein-metal complexes.

Protein-metal binding can significantly affect the passivation behavior of the alloy. Complexation of the dissolved metal ions with proteins inhibits the formation of the protective oxide. Transport of the protein-metal complexes away from the alloy/solution interface enhances further dissolution [32,33]. This phenomenon has a direct clinical relevance because protein-metal complexes can cause allergic reactions, which trigger a cell-mediated immunological response [17].

Protein binding to metal atoms in metal oxides and detachment of the protein-metal complexes may increase the dissolution rate of the alloy, however, these processes are relatively slow, and might be overlooked in the short-term studies [17]. Thus, there is a clear need for the long-term evaluation of the behavior of metal/protein systems. Additionally, the study of the kinetics of metal release in protein-containing solutions is of importance as the amount of released metals might be significantly underestimated if the complexation with proteins and precipitation of these complexes is not taken into account. Since these two processes involve non-electrochemical mechanisms, electrochemical behavior of the studied metal/protein-containing system does not always provide an adequate estimation of the metal release [26].

## 6. Albumin Precipitation

Albumin precipitation manifests itself in the formation of proteinaceous deposits on the metal surface after protein-metal binding takes place in the solution. The precipitation phenomenon on CoCrMo surfaces was reported by Ref. [27,54] after exposure to BSA-containing solutions at OCP conditions, and especially thick proteinaceous films were reported by Ref. [55,56,57,101] after potentiodynamic polarization tests. These films are particularly interesting as they resemble the carbonaceous layer found on articulating surfaces of CoCrMo hip implants [4] and are observed to be dependent on solution chemistry, pH, protein concentration and potential.

In Ref. [55], it is shown that the deposition of a thick BSA layer on the CoCrMo surface is strongly associated with the presence of Mo (VI), presumably Molybdate (MoO_4_^−2^) ions. The dedicated QCM tests on pure alloying elements of CoCrMo alloy show a significant mass increase on molybdenum samples compared to pure chromium and cobalt. Based on the XPS results, the precipitated layer was observed to be mainly composed of organic species with a minor fraction of a metallic component.

It is important to mention that protein precipitation in the form of thick films is typically observed between a potential range of 0.7 V_SCE_ and 0.8 V_SCE_ [55,56,57], which corresponds to the transpassive dissolution of a CoCrMo alloy. In Ref. [56], this potential effect is attributed to the fact that a certain amount of the released metal ions is required for the film formation.

Additionally, it is shown that protein concentration may play a crucial role in the film formation, as the potentiodynamic tests show the decrease of current in the region of transpassive potentials at 15, 30 and 45 g/L of BSA, which is attributed to the possible formation of protein film, while this effect is not observed at 60 g/L [56].

Regarding the effect of pH, it also is found that at 0.81V_SCE_ and a slightly acidic pH of 6.0, relatively thicker and more stable films than that at pH 7.6 are formed [57]. This result suggests that a local drop of pH at the electrode surface that occurs during the corrosion reaction is necessary for precipitation of the films. According to Ref. [57], the formation of the proteinaceous film may occur when the proteins are sufficiently protonated to react with the negatively charged molybdates.

The tendency of BSA to precipitate during anodization on a CoCrMo surface also is reported in cell culture medium solutions containing hyaluronic acid (HA) [101]. Then, BSA forms complexes with HA and precipitate on the surface, which is suggested to be a result of a local decrease in pH.

Precipitation of Mo-BSA complexes on a CoCrMo alloy after 14 weeks (~2352 h) of immersion tests in SBF at OCP also is reported after XPS analysis of the surfaces [54]. Contrary to these results, no signs of precipitation of Mo-protein complexes were observed in Ref. [27], in a PBS+BSA solution in the time range of 2–720 h. The decrease in the non-precipitated amount of Co released per surface area after one week of exposure is attributed to the precipitation of the Co-BSA aggregates [27].

Thus, the protein precipitation is dependent on the solution chemistry, protein concentration, solution pH, exposure time, and applied potential. However, the exact mechanism of the interaction of proteins with the released metal ions that leads to the formation of proteinaceous deposits and thick films needs further verification. It also remains unclear which protein constituents are responsible for the bonding interaction with metal ions. Regarding the formation of thick proteinaceous films, the understanding of the mechanical film integrity as the function of applied potential, pH and protein concentration can be potentially useful to define the optimum conditions which could possibly improve the corrosion kinetics and tribological behavior of the metallic implants.

## 7. Discussion

This literature overview attempts to present the current understanding of the interaction of serum albumin (both BSA and HSA) with biomedical alloys, focusing on the studies which contain mechanistic descriptions of this process supported by experimental evidences. The main attention also was placed on those studies carried out on CoCrMo, Titanium alloys and 316L steel, which are widely used biomaterials nowadays. Based on the results of this work, it can be concluded that all the albumin-related phenomena that occur during the interaction of albumin with biomedical alloys can be rationalized with four reaction mechanisms: adsorption, reduction, binding, and precipitation.

Albumin adsorption phenomena manifests itself in the adhesion of BSA molecules to the metal surface, resulting in formation of a monolayer or multilayer film. This phenomenon was reported in all the materials considered for this work. Protein reduction can be induced electrochemically and results in the reformation of the disulfide bonds of an albumin molecule. Protein binding results in the complexation of protein with the metallic species. Albumin precipitation results in precipitation of the organometallic complexes from the solution when their solubility limit is reached. So far, this phenomenon was observed in a CoCrMo alloy.

The considered albumin-related phenomena might be interrelated. Albumin adsorption, for instance, might be a preliminary step in albumin binding (complexation) with the surface oxide atoms. Albumin complexation with molybdenum leads to the precipitation of thick proteinaceous films on the surface. Albumin reduction favors the adsorption phenomena.

Figure 9 presents the summary of the parameters, which affect each of the described albumin-related phenomena. The parameters are separated into material- (green), solution- (red) and protein-related (blue). The presented parameters might be interdependent, which makes it challenging to study them separately. Surface roughness, for example, affects the wettability and surface energy, thus, the final BSA adsorption. Additionally, however, the ionic strength of the solution, pH, surface energy, chemical composition and wettability affect the surface charge. Besides material-, solution- and protein-related parameters, all the phenomena are affected by the prevailing electrochemical conditions, i.e., potential. According to the literature, the parametric dependencies are not established yet for every parameter. Additionally, the existing list of parameters in Figure 9 might not be complete yet. It remains unclear, for instance, if the protein and surface charges affect the binding phenomena or if temperature affects the precipitation phenomena.

Electrochemical potential is an important parameter affecting the interaction of albumin with biomedical alloys. A tentative summary of the literature-reported albumin-related phenomena with respect to the potential domain for 316L, CoCrMo and Ti, and Ti-6Al-4V alloys is shown in Figure 5. The data presented in this figure were selected based on the literature that contains a direct proof of the observed phenomena, while speculative references were excluded. Regarding the protein adsorption phenomena, for instance, evidences obtained with EIS, CV, QCM [24,29,39,42,43,44,46,50,67,68,69,70] and surface analysis techniques [26,29,30,34,44,46,67] were considered as sufficient. Concerning protein-metal binding phenomena, literature containing ICP-OES and AAS [26,27,51,54] evidences was considered. QCM [55,57] and EIS [56] evidences were considered for protein precipitation. Considering the protein reduction phenomena, the protein reduction on CoCrMo was not directly confirmed in Ref. [34], however this work considered the evidences of protein reduction on Hg obtained by CV and coulometry in Ref. [49]. According to Figure 5, four different potential domains can be distinguished: cathodic, OCP, passive, and transpassive. Cathodic domain implies the potentials below the corrosion potential, while passive domain corresponds to the potentials above the corrosion potential at which current remains low and relatively stable. The transpassive domain corresponds to the potentials above the passive domain and is characterized by an increase in the current density caused by the oxidation of the passive layer, its localized corrosion, or the oxidation of the environment. The OCP domain corresponds to the conditions when no current is flowing through the system and it typically lies within the passive domain of the considered biomedical alloys.

Shown in Figure 10, depending on the potential domain, different albumin-related phenomena or their combinations can occur during the interaction of biomedical alloys with albumin. Additionally, the potential does not affect the mechanisms in the same way on all the considered materials. Figure 10 clearly shows that the electrochemical conditions play a crucial role in the interaction of albumin with biomedical alloys.

Thus, the existing literature shows that protein interaction with biomedical alloys is a multidimensional phenomenon, which can involve multiple processes occurring simultaneously and also being affected by several parameters. Despite the complexity, significant progress in the field has been achieved so far. Future efforts should be dedicated to understanding the kinetics of albumin-related phenomena (and their possible combinations) that occur during interaction of albumin with biomedical alloys and development of the reliable long-term predictive models. Moreover, this understanding also will allow us to tailor surfaces for specific biomedical applications, as already attempted in the literature [102,103,104].

## 8. Conclusions

Here, a literature overview on the interaction of albumin with biomedical alloys, namely 316L steel, CoCrMo and Titanium alloys was performed, primarily focusing on the studies that presented a mechanistic description of the involved phenomena. The main outcomes of this overview are as follows:Interaction of BSA with biomedical alloys can be rationalized with four reaction mechanisms: adsorption, reduction, protein-metal binding and precipitation occurring at the surface or in the solution respectively.Electrochemical conditions have a crucial impact on the occurrence of the described mechanisms during interaction of albumin with biomedical alloys because those mechanisms depend on the potential established between the surface and the electrolyte.Parameters of material, solution, and protein, which affect each of the albumin-related mechanisms, were identified and summarized.In-situ (mainly electrochemical) and ex-situ experimental techniques used to study each of the albumin-related phenomena and their outcomes were summarized. Limitations of the experimental techniques were discussed.

It is believed that this literature overview provides a basis for further studies of the interaction of albumin with biomedical alloys in biological environments, including studies that aim at understanding the corrosion and tribological behavior of these alloys.

## Figures and Tables

**Figure 1 materials-13-04858-f001:**
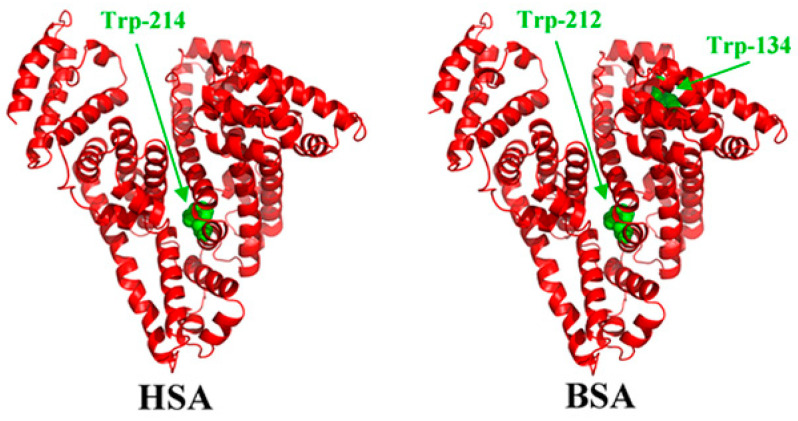
Three-dimensional structures of human serum albumin (HSA) and bovine serum albumin (BSA) with tryptophan residues in green color [25].

**Figure 2 materials-13-04858-f002:**
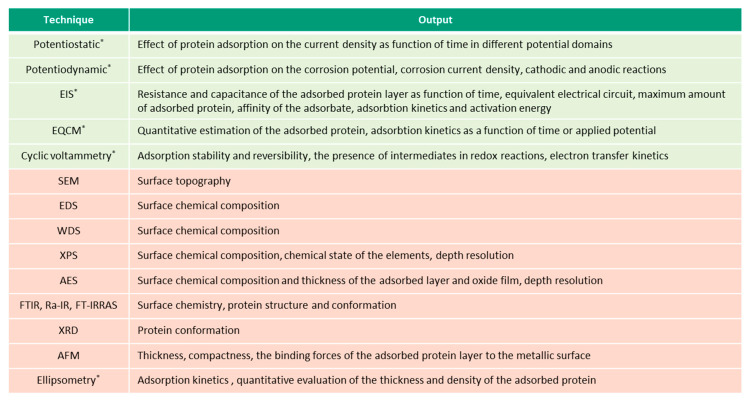
Techniques (in green—electrochemical, in-red—microscopic and spectroscopic) used for investigation of albumin adsorption on biomedical alloys, (* in-situ techniques).

**Figure 3 materials-13-04858-f003:**
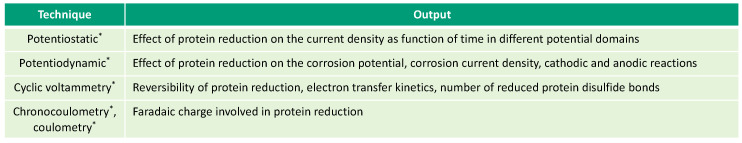
Electrochemical techniques used for investigation of albumin reduction (* in-situ techniques).

**Figure 4 materials-13-04858-f004:**
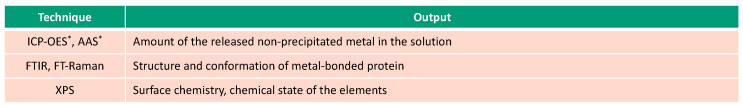
Spectroscopic techniques used for investigation of albumin-metal binding (* in-situ techniques).

**Figure 5 materials-13-04858-f005:**
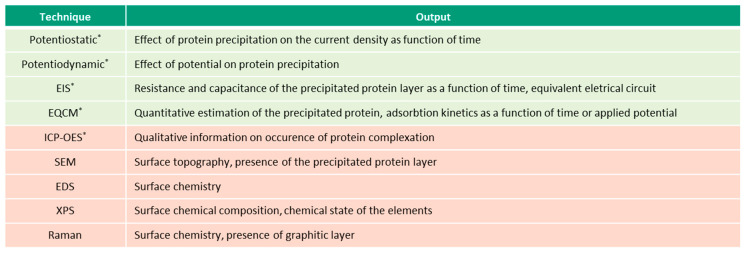
Techniques (in green—electrochemical, in-red—microscopic and spectroscopic) used for investigation of albumin precipitation (* in-situ techniques).

**Figure 6 materials-13-04858-f006:**
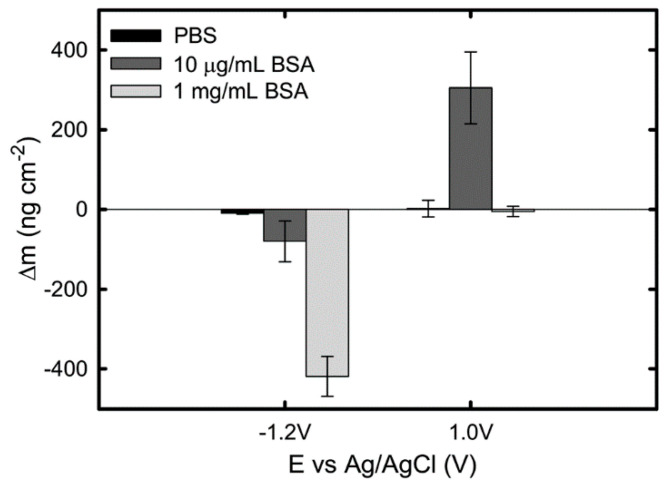
Total change of mass obtained on a pure Ti QCM sensor at cathodic and anodic potentials in PBS, PBS + 10 μg/mL BSA, and PBS + 1 mg/mL BSA solutions. Data points represent mean values of three measurements, with the standard error [39].

**Figure 7 materials-13-04858-f007:**
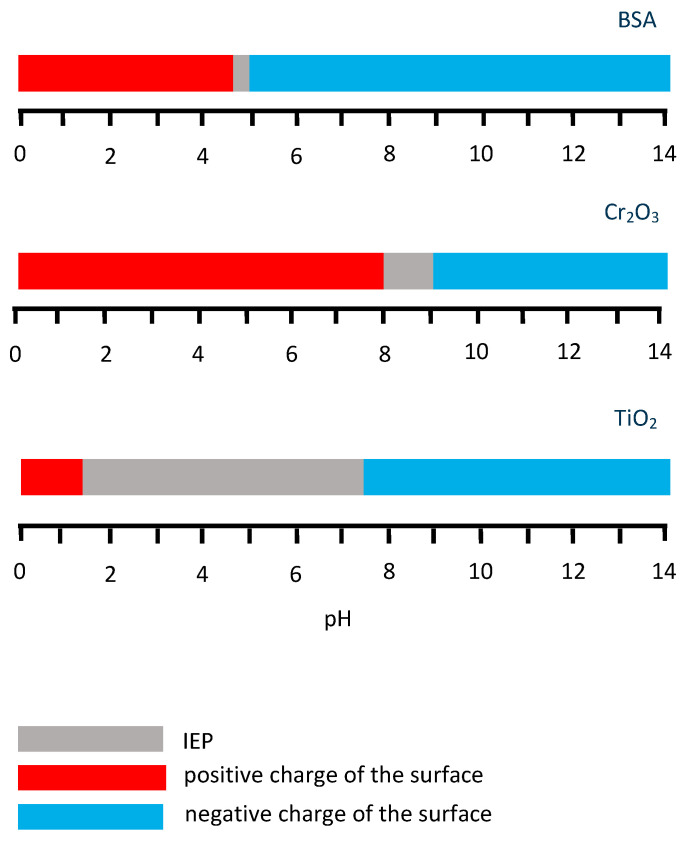
IEP ranges and surface charge of the BSA, Cr_2_O_3_ and TiO_2_ oxides.

**Figure 8 materials-13-04858-f008:**
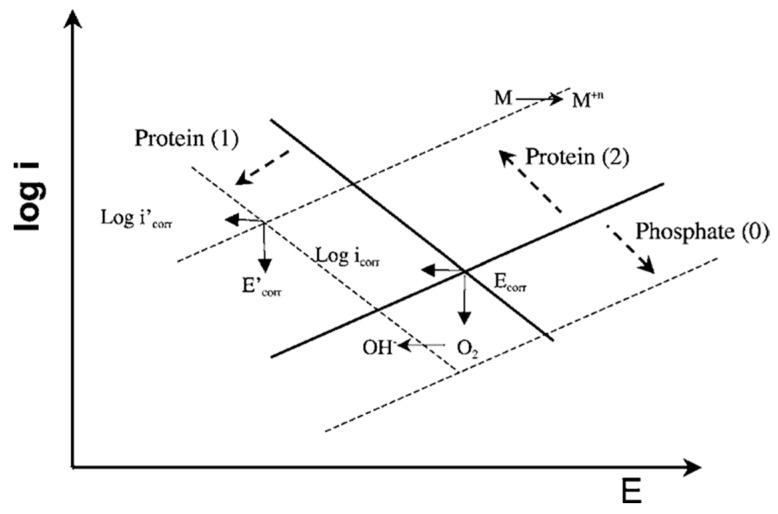
Mixed potential theory for interpretation of the effect of albumin and phosphates on the electrochemical behavior of a CoCrMo alloy. Phosphates inhibit anodic reaction (arrow 0). The albumin inhibits cathodic reaction (arrow 1) and accelerates the metal anodic oxidation by binding to metal ions (arrow 2) [29].

**Figure 9 materials-13-04858-f009:**
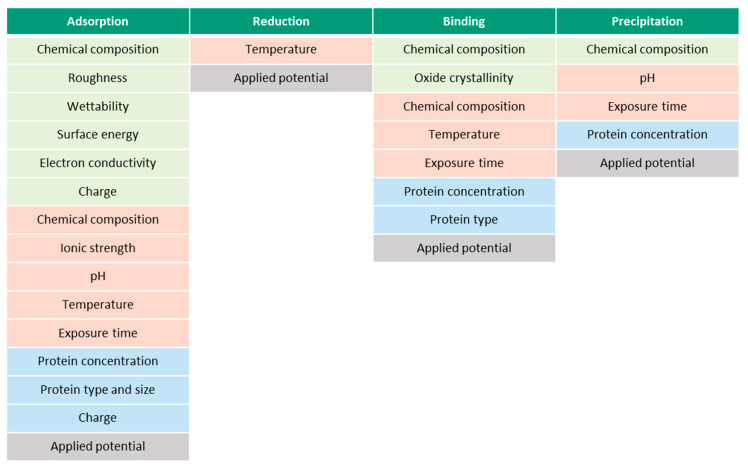
Effect of various parameters (material- (green), solution- (red) and protein-related (blue)) on protein-related phenomena.

**Figure 10 materials-13-04858-f010:**
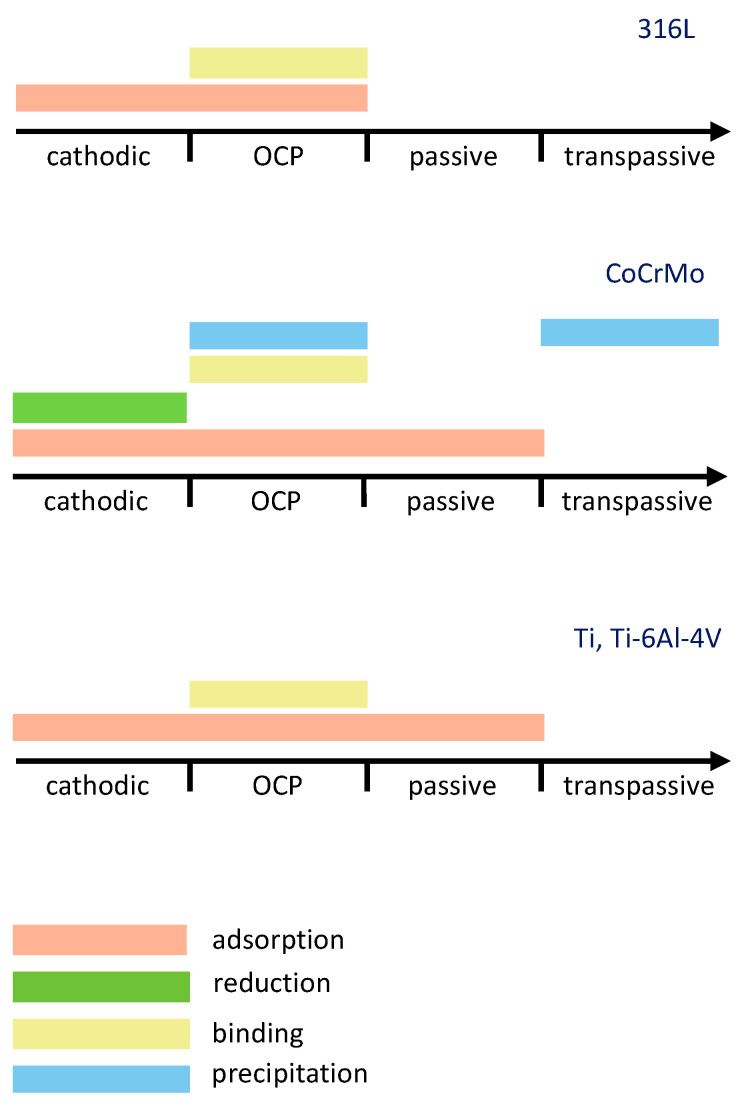
Effect of electrochemical potential on the occurrence of albumin-related phenomena as a result of interaction of albumin with 316L steel, CoCrMo, Ti and Ti-6Al-4V alloys.

**Table 1 materials-13-04858-t001:** Summary of the published papers containing mechanistic studies on the interaction of serum albumin with biomedical alloys.

Author	Year	Material	Material	Electrochemical Technique	Characterization	Phenomenon
[15]	2013	CoCrMo, Ti, Al etc.				Adsorption
[16]	2020	CoCrMo	PBS ^a^ + BSA	QCM ^f^	XRD ^l^, AES ^m^, XPS ^n^, AFM ^o^	Adsorption
[24]	2010	CoCrMo	0.14 M NaCl + BSA	OCP ^g^, Potentiodynamic, EIS ^h^		Adsorption
[29]	2007	CoCrMo	PBS + BSA, NaCl, NaCl + BSA	OCP, Potentiostatic, Potentiodynamic, EIS	AES, XPS	Adsorption
[30]	2012	CoCrMo	NaCl + BSA	OCP, Potentiostatic, Potentiodynamic, EIS, EQCM ^i^	XPS	Adsorption
[31]	2011	CoCrMo	PBS + BSA	OCP, Potentiodynamic, EIS		Adsorption
[39]	2019	Ti	PBS + BSA	Potentiostatic, Potentiodynamic, EQCM	AFM	Adsorption
[40]	2008	316L, CoCrMo	NaCl + BSA, PBS+BSA	OCP, Potentiodynamic, EIS		Adsorption
[41]	2007	Cr	0.05 M Na_2_SO_4_ + BSA	EQCM	XPS	Adsorption
[42]	2000	Ti	PBS + BSA	CV ^j^, EIS		Adsorption
[43]	2009	TiO_2_	0.1 M NaCl + HSA	CV, EIS		Adsorption
[44]	2020	CoCrMo	PBS + BSA	OCP, EIS	SEM ^p^, AFM, EDS ^q^, FIB ^r^/TEM ^s^	Adsorption
[45]	2015	CoCrMo	PBS + BSA	OCP, Potentiodynamic, Potentiostatic	AFM, AES	Adsorption
[46]	2019	CoCrMo	acidic artificial saliva + BSA	OCP, EIS	SEM, EDS, XRD, FT-IR ^t^, RA-IR ^u^	Adsorption
[47]	2010	Ti-Cr, Ti-Al, Ti-Ni, Al-Ta, Al-Zr	PBS + BSA, PBS + Fbn ^b^		WDS ^v^, XRD, XPS	Adsorption
[48]	2003	Cr, Mo	PBS + BSA, H_2_O + BSA		XPS, FT-IRRAS ^w^	Adsorption
[34]	2015	CoCrMo	0.14 M NaCl + BSA	Potentiodynamic, EQCM	XPS	Adsorption, Reduction
[49]	1978	Hg	PBS+BSA	CV, Chronocoulometry		Reduction
[28]	2018	316L	0.9wt% NaCl + BSA	OCP, Immersion tests, Polarisation, EIS	XPS	Adsorption, Binding
[50]	2012	316L, 430, 304, LDX 2205, Cr	PBS + BSA, PBS + HSA, PBS + LYZ ^c^, PBS + BSM ^d^	QCM	AFM, XPS, AAS ^x^	Adsorption, Binding
[51]	2014	CoCrMo, Ti–6Al–4V, 316L	PBS + HSA	Immersion tests	XPS, ICP-OES ^y^	Adsorption, Binding
[52]	2013	316L	PBS + BSA, PBS + LYZ	QCM, OCP, LPR ^k^	AAS, XPS	Adsorption, Binding
[26]	2017	316L	PBS + BSA, PBS + BSA + Fbn	OCP, Polarization resistance	AAS, XPS	Binding
[27]	2014	CoCrMo, 316L	PBS + BSA	OCP	AAS, XPS	Binding
[53]	2008	V species	Tris–HCl + BSA		FT-IR, FT-Raman, UV–Vis ^z^	Binding
[54]	2012	CoCrMo, Ti–6Al–4V, 316L	PBS + BSA	Immersion tests	XPS, ICP-OES	Adsorption, Binding, Precipitation
[55]	2013	CoCrMo	BCS ^e^	Potentiodynamic, EQCM	XPS	Precipitation
[56]	2017	CoCrMo	BCS	Potentiostatic, Potentiodynamic, EIS	SEM, Raman	Precipitation
[57]	2015	Cr	Tris buffer+BSA	QCM		Precipitation

^a^ PBS—phosphate buffer solution, ^b^ Fbn—fibrinogen, ^c^ LYZ—lysozyme, ^d^ BSM—bovine serum mucin, ^e^ BCS—bovine calf serum, ^f^ QCM—quartz crystal microbalance, ^g^ OCP—open circuit potential, ^h^ EIS—electrochemical impedance spectroscopy, ^i^ EQCM—electrochemical quartz crystal microbalance, ^j^ CV—cyclic voltammetry, ^k^ LPR—linear polarization resistance, ^l^ XRD—X-ray diffraction, ^m^ AES—Auger electron spectroscopy, ^n^ XPS—X-ray photoelectron spectroscopy, ^o^ AFM—atomic force microscopy, ^p^ SEM—scanning electron microscopy, ^q^ EDS—energy-dispersive X-ray spectroscopy, ^r^ FIB—focused ion beam, ^s^ TEM—transmission electron microscopy, ^t^ FT-IR—Fourier-transform infrared spectroscopy, ^u^ RA-IR—reflection absorption infrared spectroscopy, ^v^ WDS—wavelength-dispersive spectroscopy, ^w^ FT-IRRAS—Fourier-transform infrared reflection−absorption spectroscopy, ^x^ AAS—atomic absorption spectrometry, ^y^ ICP-OES—inductively coupled plasma optical emission spectrometry, ^z^ UV-Vis—ultraviolet-visible.

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
