# Peer review of "An Overview of Serum Albumin Interactions with Biomedical Alloys"

_materials, 2020, doi:10.3390/ma13214858_

Round 1

Reviewer 1 Report

This is a very interesting work, that fits well with the scope of this journal. The manuscript is well written and organized, literature analysis is sufficient and a consisted and interesting discussion is included. Summary graphs in discussion are very illustrative and provide a nice overview of the focus of this work.  My only comment/remark is that different techniques are used to monitor similar phenomena, so it would be nice to add a comment on how these methods complement each other and (if you have any information) on their repeatability and reproducibility.   

Reviewer 2 Report

Dear Authors,

Congratulations on a very well written article. I am very impressed with the description of tribocorrosion and reviewing a large amount of data and comparison - especially Table 1 (Summary of the published papers containing mechanistic studies on the interaction of serum albumin with biomedical alloys).

The only thing I can refer to which can be improved and which will certainly have a positive impact on the quoted your article in the future is to refer to newer sources in the field of electrochemical research because that is what you lack. Especially focus on EIS because there are a lot of newer papers s in this area. Read the following ones, refer to them and quote them because they are really cool work:

  • Effect of thin SiO2 layers deposited by means of atomic layer deposition method on the mechanical and physical properties of stainless steel | Einfluss dünner SiO2-Schichten, die mittels Atomlagenabscheidung aufgebracht wurden, auf die mechanischen und physikalischen Eigenschaften von Edelstahl,
  • The development of an innovative nano-coating on the surgical 316 L SS implant and studying the enhancement of corrosion resistance by electrochemical methods using Ibandronate drug
  • EIS comparative study and critical Equivalent Electrical Circuit (EEC) analysis of the native oxide layer of additive manufactured and wrought 316L stainless steel

Best Wishes,

Realy Good Job :)

Reviewer
